# Revisiting the Role of Autophagy in Cardiac Differentiation: A Comprehensive Review of Interplay with Other Signaling Pathways

**DOI:** 10.3390/genes14071328

**Published:** 2023-06-24

**Authors:** Mina Kolahdouzmohammadi, Roya Kolahdouz-Mohammadi, Seyed Abdolhossein Tabatabaei, Brunella Franco, Mehdi Totonchi

**Affiliations:** 1Department of Genetics, Reproductive Biomedicine Research Center, Royan Institute for Reproductive Biomedicine, ACECR, Tehran P.O. Box 16635-148, Iran; 2Department of Nutrition, School of Public Health, Iran University of Medical Sciences, Tehran 1449614535, Iran; 3Shariati Hospital, Faculty of Medicine, Tehran University of Medical Sciences, Tehran 1411713135, Iran; 4Telethon Institute of Genetics and Medicine (TIGEM), Via Campi Flegrei, 34, 80078 Pozzuoli, Italy; 5Genomics and Experimental Medicine Program, Scuola Superiore Meridionale (SSM, School of Advanced Studies), 80138 Naples, Italy; 6Medical Genetics, Department of Translational Medicine, University of Naples “Federico II”, Via Sergio Pansini, 80131 Naples, Italy; 7Department of Environmental, Biological and Pharmaceutical Sciences and Technologies (DiSTABiF), Università degli Studi della Campania “Luigi Vanvitelli”, 81100 Caserta, Italy

**Keywords:** autophagy, cardiomyocyte, differentiation

## Abstract

Autophagy is a critical biological process in which cytoplasmic components are sequestered in autophagosomes and degraded in lysosomes. This highly conserved pathway controls intracellular recycling and is required for cellular homeostasis, as well as the correct functioning of a variety of cellular differentiation programs, including cardiomyocyte differentiation. By decreasing oxidative stress and promoting energy balance, autophagy is triggered during differentiation to carry out essential cellular remodeling, such as protein turnover and lysosomal degradation of organelles. When it comes to controlling cardiac differentiation, the crosstalk between autophagy and other signaling networks such as fibroblast growth factor (FGF), Wnt, Notch, and bone morphogenetic proteins (BMPs) is essential, yet the interaction between autophagy and epigenetic controls remains poorly understood. Numerous studies have shown that modulating autophagy and precisely regulating it can improve cardiac differentiation, which can serve as a viable strategy for generating mature cardiac cells. These findings suggest that autophagy should be studied further during cardiac differentiation. The purpose of this review article is not only to discuss the relationship between autophagy and other signaling pathways that are active during the differentiation of cardiomyocytes but also to highlight the importance of manipulating autophagy to produce fully mature cardiomyocytes, which is a tough challenge.

## 1. Introduction

The process of cardiomyogenesis, the transformation of stem cells into functional cardiomyocytes, is a complex and precisely organized series of committed differentiation steps that involve the mesoderm, cardiac mesoderm, and cardiac progenitors [1]. Autophagy, a cellular process that degrades cytosolic proteins and subcellular organelles in lysosomes, plays a key role in the differentiation of various cell types, including primary cells and stem cells [2,3,4,5,6,7,8]. 

The involvement of autophagy has also been extensively studied and demonstrated in cardiomyocyte differentiation [9,10,11,12]. Autophagy, triggered by energy or nutrient deprivation or as a quality-control mechanism, is crucial for cellular homeostasis and survival by providing cells with building blocks and energy for cellular reconstruction and maintenance. Moreover, autophagy regulates a variety of critical cellular processes including self-renewal, differentiation, senescence, and apoptosis [11,13,14]. In particular, autophagy is activated during differentiation to perform necessary cellular remodeling, including protein turnover, lysosomal degradation of organelles, and to ensure the quality of intracellular proteins and organelles [10,15]. Autophagy has been found to play a significant role in regulating cardiomyocyte differentiation by degrading and recycling damaged or unneeded components. Reducing oxidative stress and promoting energy homeostasis are two additional ways in which autophagy has been proven to positively affect the survival and function of differentiated cardiomyocytes. In general, autophagy and cardiac differentiation are closely interrelated processes that work together to produce fully mature cardiomyocytes [9,16,17].

In addition to autophagy, mitophagy, the selective degradation of dysfunctional or damaged mitochondria, plays a crucial role in cardiomyocyte differentiation by orchestrating a series of cellular processes. Mitochondria play an important role as key regulators in this developmental process because of their ability to respond to nutrient signals, dynamically adjust their respiratory capacity, and control the production of reactive oxygen species (ROS). Mitophagy is a process that allows cardiomyocytes to selectively eliminate dysfunctional mitochondria, ensuring the maintenance of a healthy mitochondrial population. This active mitochondrial quality-control mechanism not only protects the cell from the deleterious effects of damaged mitochondria but also promotes the initiation of transcriptional and epigenetic changes that direct the maturation of cardiomyocytes [18].

Considering the benefits of regulating autophagy during the development of cardiomyocytes such as (but not limited to) improved cellular homeostasis, better control of protein quality, reduced oxidative stress, enhanced cell survival, and lower amount of pathological remodeling [1,19,20,21,22], it becomes important to find out how modifying autophagy affects cardiomyocyte differentiation. Additionally, establishing a reliable method for generating fully mature cardiomyocytes has been a persistent challenge with obvious translational implications and autophagy modification may provide a potential solution. Therefore, the objective of this article is to summarize and review our knowledge in this field, aiming at determining whether altering the autophagy pathway can enhance the differentiation process of cardiomyocytes and address this challenge.

## 2. Crosstalk between Autophagy, Cardiac Differentiation, and Other Signaling Pathways

The crosstalk between autophagy and other signaling pathways plays a crucial role in the modulation of cardiac differentiation. Zhang et al. were the first to recognize the connection between autophagy and cardiac stem-cell differentiation, finding that fibroblast growth factor (FGF) regulates cardiac development through autophagy [8]. Their study revealed that the inhibition of FGF signaling led to an upsurge in the initiation of autophagy, thereby promoting the differentiation of cardiomyocytes. These findings strongly suggest that autophagy serves as a positive regulator of both cardiac progenitor cell differentiation and cardiomyocyte maturation.

FGF signaling exhibits a biphasic function in cardiogenesis. During the early stages, FGF signaling promotes mesoderm formation and commitment to the cardiac lineage. However, in later stages, FGF signaling inhibits premature differentiation to ensure proper cardiac development and maturation. The involvement of autophagy in mediating FGF signaling during the differentiation of heart progenitor cells highlights the significance of autophagy in cardiac development. The interplay between autophagy and FGF signaling provides novel insights into the regulatory mechanisms underlying cardiac differentiation. Moreover, FGF signaling has been found to inhibit autophagy activity, while the suppression of FGF signaling increases autophagy initiation [8,10]. This further supports the intricate relationship between FGF and autophagy in cardiac differentiation.

Additionally, other signaling pathways, such as Wnt, Notch, and bone morphogenetic proteins (BMPs), are also involved in regulating heart development [23]. Wnt and Notch signaling pathways are important in heart development, and there is evidence of a crosstalk between the two [24]. The main components of the Wnt signaling pathway include dishevelled (Dvl), AXIN, glycogen synthase kinase-3 (GSK3)-β, and β-catenin. It has been shown that Wnt signaling plays a biphasic function in cardiogenesis. This pathway stimulates cardiomyocyte differentiation in the early stages of heart development but suppresses cardiogenesis later on [25]. Autophagy can be negatively regulated by Wnt signaling by promoting Dvl degradation through the autophagy–lysosome pathway, which, in turn, induces a concomitant decrease in the nuclear accumulation of β-catenin [26]. Furthermore, it has been demonstrated that autophagy induction resulted in a considerable drop in the gene expression of *Axin2*, *Cyclin-D1*, and *c-Myc*, as well as a decrease in the protein expression of β-catenin and Dvl2. Based on these findings, it was determined that autophagy has a negative regulatory effect on the Wnt signaling pathway [1]. Moreover, GSK3, a component of the β-catenin-WNT pathway, directly regulates autophagy through the GSK3-KAT5/TIP60-Unc-51 like autophagy activating kinase 1 (ULK1) signal [27]. Additionally, the differentiation of cardiac stem cells (CSCs) into cardiomyocytes is influenced by the Jun N-terminal kinase (JNK)/signal transducer and activator of transcription 3 (STAT3) and β-catenin pathways [28,29,30,31].

Notch signaling, a pivotal signaling pathway in cardiac differentiation, demonstrates intricate dynamics and complexity throughout the process of cardiac development. Initially, it is believed that Notch signaling suppresses cardiac differentiation to maintain a pool of undifferentiated progenitor cells. However, different studies show that Notch1 is required for cardiac progenitor cells to differentiate into cardiomyocytes by positively regulating the expression of cardiac transcription factors in mouse embryos and embryonic stem (ES) cells [32,33,34]. In the context of cardiac development, autophagy has been investigated in relation to the Wnt and Notch signaling pathways [35,36]. Zhuqing Jia et al. studied the involvement of autophagy in the cardiac development of P19CL6 cells, a well-established in vitro cardiomyocyte differentiation system [37]. Their findings demonstrated that autophagy is initiated during the early stages of cardiomyocyte differentiation and remains active throughout the final stages. The researchers also observed that β-catenin and Notch intracellular domain (NICD), which are effectors of the Wnt and Notch pathways, can form a complex with LC3 and P62, leading to their elimination through autophagy. Interestingly, β-catenin not only inhibits the formation of autophagosomes but also directly reduces the expression of the autophagy adaptor p62 [36]. Since both the Notch and Wnt pathways are inhibitory in the late stages of cardiac differentiation, appropriate clearance of both pathways’ components may allow P19CL6 cells to complete the cardiac differentiation process [37,38].

Moreover, epigenetic factors, such as histone deacetylases (HDACs), have also been shown to play a role in regulating autophagy during cardiac differentiation [27,28,29,30,31]. Specifically, studies have demonstrated that HDACs control the differentiation of P19 embryonic cancer cells into cardiac lineage cells [39]. Despite the advances in understanding the roles of autophagy and epigenetic regulation in cardiac differentiation, the exact mechanisms underlying their interaction in this process remain incompletely understood. Further investigations are required to elucidate the intricate relationship between autophagy, Wnt signaling, and cardiac differentiation, which will provide valuable insights into the development and potential therapeutic strategies for cardiac-related conditions.

The mammalian target of rapamycin (mTOR) is a crucial serine/threonine kinase involved in cellular survival and growth. It operates through two distinct complexes, mTOR complex 1 (mTORC1) and mTOR complex 2 (mTORC2), which are predominantly made up of Raptor and Rictor as their core components, respectively. Reports from different laboratories have demonstrated that mTOR complex 1 acts as a master regulator of autophagic processes since inhibition of mTORC1 was shown to be required to start autophagy. Subsequent studies have also demonstrated that mTORC1 directly regulates the subsequent phases, such as autophagosomes formation, maturation, and autophagy termination [40]. Raptor/mTORC1 and Rictor/mTORC2 play distinct and significant roles in the differentiation of cardiomyocytes derived from mouse embryonic stem cells (mESCs). Numerous molecular elements and phosphorylation processes mediate the intricate interaction between these two signaling pathways [41].

AKT, also known as protein kinase B, is a key protein and acts as a bridge between the two complexes. AKT can be phosphorylated by mTORC2 or 3-phosphoinositide-dependent kinase 1 (PDK1), thereby regulating downstream signaling [42]. Specifically, Raptor knockdown reduced the phosphorylation of Rictor at Thr1135 by p70S6K, leading to mTORC2 activation [41].

Raptor/mTORC1 primarily regulates metabolism, cell growth, and protein synthesis in response to nutrient availability, and growth factors [43]. It controls the phosphorylation of S6 kinase 1 (S6K1) and eukaryotic translation initiation factor 4E-binding protein (eIF4E), thereby promoting translation and protein synthesis [44]. In the context of cardiomyocyte differentiation, Raptor knockdown enhances the process by activating Rictor/mTORC2 signaling, which, in turn, facilitates the differentiation of mESCs into cardiomyocytes. This is accompanied by elevated expression levels of brachyury (a mesoderm protein), Nkx2.5 (a cardiac progenitor cell protein), and α-actinin (a cardiomyocyte marker) in Raptor knockdown cells [41].

On the other hand, Rictor/mTORC2 exerts its impact on cytoskeletal organization, cell survival, and cell polarization. It has been implicated in the regulation of cellular processes such as cell adhesion, migration, and differentiation [45]. Rictor knockout in cardiac-specific cells leads to cardiac dysfunction and impaired adaptation to pressure overload. In the context of mESCs differentiation into cardiomyocytes, Rictor knockdown suppresses cardiogenesis, indicating the critical role of Rictor/mTORC2 in this process [41].

Upstream of mTOR, various signals, including insulin signaling through phosphatidylinositol-3-kinase (PI3K) and AKT, and energy/stress response signaling via AMP-activated protein kinase (AMPK), converge to regulate the activity of the tuberous sclerosis complexes (TSC1-TSC2), which act as GTPase-activating proteins for Rheb. When Rheb is in its GTP-bound state, it directly activates mTOR. The convergence of these signals ensures precise control over mTOR activation and downstream cellular processes [44].

The regulation of cardiomyocyte differentiation can be better understood by gaining valuable insights from the complex relationship between these signaling pathways, which may also have implications for potential therapeutic applications using induced pluripotent stem cells.

Aside from the main autophagy pathway, other types of autophagy, such as mitophagy, play a crucial role in cardiac differentiation. Mitophagy, a form of autophagy, is a vital process in cardiac differentiation. It helps in the removal of dysfunctional mitochondria and facilitates metabolic remodeling, which is crucial for successful differentiation. Regulated mitophagy plays a critical role in maintaining the cell’s glycolytic status and is believed to contribute to the metabolic remodeling necessary for successful differentiation [46,47,48,49].

A specific process of mitophagy is initiated by the PINK1/Parkin system, where Parkin, an E3 ubiquitin ligase, is recruited to dysfunctional mitochondria. Parkin ubiquitinates outer mitochondrial proteins, labeling them for proteasome degradation [50]. In mice, the deletion of Parkin during early postnatal stages impedes cardiomyocyte maturation, affects mitochondrial biogenesis, and disrupts fatty acid oxidation, highlighting the importance of PINK1/Parkin-mediated mitophagy in cardiac development [46]. These findings highlight the important role of PINK1/Parkin-mediated mitophagy in promoting proper cardiac development.

The regulation of mitophagy in cardiac differentiation is influenced by nutrient-sensing pathways, such as Akt/mTOR, which is upregulated in high-nutrient states, and AMPK, which is upregulated during fasting. Inhibiting mTOR signaling increases mitophagy, providing cardioprotection after myocardial infarction. Conversely, AMPKα2 phosphorylates PINK1, enhancing mitophagy and protecting against pressure overload in the heart. AMPK also influences the opening of the mitochondrial permeability transition pore (mPTP), which regulates mitochondrial depolarization and subsequent mitophagy [18,51]. Increasing Parkin-mediated mitophagy through enhanced fatty acid oxidation, achieved by deleting acetyl coenzyme A carboxylase 2 (ACC2), prevents myocardial dysfunction resulting from a high-fat diet [52]. These interconnected mechanisms underscore the significance of metabolic shifts, mitochondrial quality control, and mitophagy in maintaining proper cardiac function.

Conversely, inhibiting mitophagy by chronically activating monoamine oxidase-A (MAO-A) leads to the accumulation of dysfunctional mitochondria, promoting a senescent phenotype. This inhibition upregulates cell-cycle inhibitors, triggers the ROS-induced DNA damage response, and increases senescence-associated β-galactosidase (SA-β-gal) activity. Hence, maintaining mitophagy is crucial for inhibiting senescence, promoting cardiomyocyte maturation, removing dysfunctional mitochondria, and supporting optimal mitochondrial function [53].

The hypoxia-inducible factor 1 (HIF-1) is one of the key regulators of mitophagy during cardiac differentiation. HIF-1 is known to stabilize and activate in response to low oxygen levels, and it has been found to play a critical role in cardiomyocyte development [54]. Zhao et al. (2020) found that HIF-1 is upregulated during differentiation and triggered NIX (also known as BNIP3L)-dependent mitophagy [55]. NIX acts as a mitophagy receptor that facilitates the selective engulfment of mitochondria by autophagosomes. Activation of HIF-1 during cardiomyocyte differentiation leads to the stabilization of NIX, promoting the removal of dysfunctional mitochondria through mitophagy. This controlled elimination of damaged mitochondria is essential for maintaining the overall health and functionality of differentiating cardiomyocytes. Controlled mitophagy is thought to maintain or improve the cell’s glycolytic status, potentially through a positive feedback loop with HIF1, although the mechanism is not yet understood. Changing NIX expression levels can affect the rate of differentiation, and HIF1 upregulation during differentiation is required for cardiomyoblast development and mitophagy [55,56].

Figure 1 summarizes the data available to date and the pathways linking autophagy to cardiac differentiation.

## 3. How Does Manipulation of Autophagy Improve Cardiac Differentiation?

Recent advances in stem-cell research have utilized embryoid body suspension and monolayer attachment methods to increase differentiation efficiency, but further improvements are required to optimize this process [22,57,58]. Adding growth factors such as BMP2, BMP4, FGFs, activin A, b-fibroblast growth factor (bFGF), FGF2, FGF10, and Wnt family member 3A (Wnt3a) during cardiac differentiation can improve induction efficiency, but the high cost of these reagents limits their practicality [1,59,60]. Thus, the use of small chemical cocktails that target the autophagy pathway is a promising alternative (Figure 2). However, in order to boost differentiation efficacy even further, the induction strategy that improves differentiation efficiency, as well as the detailed mechanisms, must be defined.

Methods that have been developed to manipulate the autophagy pathway to promote cardiac differentiation are discussed below (Figure 3 and Table 1). Rapamycin is an inhibitor of mTOR that has been found to promote cardiomyocyte differentiation in a stage-dependent manner. It works similarly to Ku0063794, another mTOR kinase inhibitor [1]. While rapamycin inhibits mTORC1 activity, its effect on mTORC2 is limited [61]. Nevertheless, rapamycin has been found to induce autophagy, upregulate Fgf8 and Nodal expression, alleviate stressors during cardiac differentiation, reduce hPSC apoptosis, promote BMP signaling, and suppress the activation of Wnt/β-catenin and Notch signaling during cardiac induction [21,22,38,62,63]. The efficiency of cardiac differentiation can be affected by different rapamycin concentrations and the presence of CHIR99021, a small molecule inhibitor that targets the GSK-3β enzyme. In addition to reducing hPSC apoptosis and boosting hESC survival, rapamycin has been shown to have other benefits in cardiac differentiation. It has been found to reduce medium usage by around 50% and overcome variability in differentiation efficiency among different hESC lines and repetitions, resulting in a greater efficiency, ranging from 87% to 99.06% [22].

Moreover, combining rapamycin and ascorbic acid has also been suggested as an effective regimen for cardiac induction. This combination has been found to increase collagen IV synthesis as well as a number of molecular signals triggered by collagen, such as the extracellular signal-regulated kinase (ERK), c- JNKs, and STAT1/3 pathways, which promote cardiac differentiation through various molecular signaling pathways [66]. These findings provide insight into the diverse and complex mechanisms underlying the regulation of cellular differentiation and suggest the potential therapeutic applications of rapamycin in promoting cardiac regeneration.

AMPK is an energy-sensing protein kinase involved in the regulation of autophagy. In addition to its direct role in regulating autophagy, AMPK can also influence other cellular processes, such as mitochondrial function, posttranslational acetylation, cardiomyocyte metabolism, mitochondrial autophagy, endoplasmic reticulum stress, and apoptosis. As AMPK is involved in the control of various cellular processes, it can influence the health and survival of cardiomyocytes [20,67]. AMPK activation increases fatty acid oxidation in human induced pluripotent stem-cell (hiPSC)-cardiomyocytes (CMs) by upregulating the expression of carnitine palmitoyl transferase (CPT)-1, fatty acid transport protein (FATP), and fatty acid-binding protein (FABP) [20,68]. Moreover, AMPK has a distinct regulatory impact on numerous areas of mitochondrial biology and homeostasis, such as controlling the number of mitochondria via stimulating mitochondrial synthesis, regulating the architecture of the mitochondrial network, and controlling the quality of mitochondria via autophagy and mitosis [69].

Understanding the significance of AMPK in cardiomyocyte differentiation, Ye et al. employed 5-aminoimidazole-4-carboxamide-1-β-D-ribofuranoside (AICAR) to induce AMPK activation in hiPSC-CMs [19]. AICAR dramatically raised the oxygen consumption rate (OCR) related to baseline respiration, adenosine 5′-triphosphate (ATP) generation, maximum respiration, and reserve capacity, indicating that cardiomyocytes were in a more mature metabolic state. Furthermore, AMPK activation enhances mitochondrial fusion in hiPSC-CMs [19]. In times of cellular stress, mitochondrial respiratory reserve capacity is employed to sustain an increase in energy demand, assisting in the maintenance of cell and organ functions, cell repair, or active drug detoxification [70]. AICAR treatment boosted respiratory reserve capacity in hiPSC-CMs, implying that the treated cells may perform better in the face of increasing energy demand. In addition, AICAR-treated hiPSC-CMs exhibited a larger cell perimeter, a lower circularity index, and longer sarcomeres than the control group, making them more similar to adult cardiomyocytes [19]. AICAR also increased the expression of cardiac maturation-related genes. Although the AICAR activation of AMPK promotes the morphological and metabolic maturity of hiPSC-CMs, the effect of AMPK on the electrophysiological maturation of hiPSC-CMs remains unknown [19].

In a recent study, the impact of an AMPK inducer (Metformin) on the differentiation of hPSC-CMs was investigated. The findings demonstrated a significant upregulation of autophagy during cardiac differentiation following AMPK activation. Additionally, this autophagy upregulation was associated with an increase in the expression of CM-specific markers in hPSC-CMs. These findings provide valuable insights into the vital role of autophagy in the process of cardiomyocyte differentiation [17].

There is evidence to suggest that β-cyclodextrin (β-CD) has effects on cholesterol metabolism and autophagy activation which can aid in the differentiation of many cells, including chick cardiac cells [71,72,73]. It has been observed that modifying cholesterol metabolism induces autophagy, which activates the JNK/STAT3 and GSK3-β/β-catenin pathways, resulting in increased expression of cardiac transcription factors such as Nkx2.5 and GATA binding protein 4 (GATA4), structural proteins such as cardiac Troponin T, transcriptional enhancers such as Mef2c, and inducing GATA4 nuclear translocation [57].

## 4. Other Forms of Autophagy and Cardiac Differentiation

Mitophagy, the selective removal of damaged or depolarized mitochondria, plays a crucial role in maintaining mitochondrial health as a quality control mechanism [74]. During stem cell differentiation, changes in metabolism occur to increase oxidative phosphorylation and mitochondrial ATP production [75]. This metabolic shift has been shown to drive cell differentiation, implying a positive feedback loop between differentiation and metabolic rewiring [76].

The involvement of mitochondrial biogenesis and the β-catenin pathway in the differentiation of endothelial progenitor cells is supported by scientific evidence. An increase in mitochondrial biogenesis, associated with the upregulation of PGC-1α, has been observed during differentiation towards the endothelial progenitor cell stage [65,77]. PGAM5, a mitochondrial phosphatase released during mitophagy, activates the β-catenin pathway by dephosphorylating β-catenin, allowing it to translocate to the nucleus and transcribe Wnt/β-catenin pathway genes and promoting mitophagy and mitochondrial biogenesis [78,79]. This interplay between PGAM5 and β-catenin is crucial for the differentiation of hiPSCs into endothelial cells [65].

While the downstream impact of mitophagy on iPSC differentiation has been studied, the upstream initiators of mitophagy during cell differentiation remain unknown. Krantz et al. investigated the role of mitophagy in iPSC differentiation and the initiation of compensatory mitochondrial biogenesis via the PGAM5 pathway but did not identify the upstream initiators of mitophagy [65]. In the review article authored by Garbern in 2021, a comprehensive examination of the role of mitophagy in cardiomyocyte differentiation is presented [18].

## 5. Discussion

The aim of this review was to explore and understand the interplay between cardiac differentiation signaling pathways and autophagy, and to evaluate the significance of autophagy in the differentiation of cardiomyocytes. Autophagy is critical for cell homeostasis and has a dual role in both cell survival and death during stressful conditions like differentiation, and its regulation is crucial to avoid harmful effects [80]. Understanding the interplay between autophagy and cardiac differentiation signaling pathways may provide insights into the process of generating cardiomyocytes with more mature characteristics.

Multiple studies have explored the modulation of autophagy during cardiomyocyte differentiation in an effort to optimize the differentiation protocol. The majority of these studies have focused on the manipulation of the mTOR pathway using rapamycin, but there is still much uncertainty due to the complex nature of the autophagy pathway [1,19,20,21,22,55]. To address these uncertainties, future studies should use multiple small molecules to induce/inhibit autophagy and examine their role in heart differentiation.

Although the role of autophagy during cardiac differentiation has been partially answered, one of the critical unanswered questions is the precise mechanism by which mitophagy contributes to cardiomyocyte differentiation. Studies have suggested that differentiation-inducing growth factors may be potential initiators of mitophagy, but other environmental signals such as matrix signaling, cell–cell interaction, or mechanosignaling may also play a role [65]. Understanding the exact cues that initiate mitophagy during cardiac differentiation is critical for developing effective strategies to promote cardiomyocyte maturation. Manipulation of the expression or activity of proteins involved in mitophagy, for example, could potentially improve the removal of damaged mitochondria and improve cardiomyocyte function [81,82]. The mechanisms underlying the regulation of mitophagy during cardiac differentiation, however, are largely unknown, and more research is needed to elucidate the signaling pathways and molecular mechanisms involved. Finally, gaining a better understanding of how mitophagy is regulated during cardiac differentiation could have significant implications for the generation of mature cardiomyocytes.

Our knowledge concerning the function of autophagy in the path of cardiomyocyte differentiation is currently ambiguous due to some limitations. For example, gene silencing and overexpression of autophagy pathway genes have not been examined to modify autophagy levels in PSCs during cardiomyocyte differentiation. Furthermore, other forms of autophagy, such as chaperone-mediated autophagy and microautophagy, must be studied for their potential role in cardiomyocyte differentiation. More research is required to better understand these complex processes and their role in cardiac differentiation, which may lead to the development of more effective strategies for generating mature cardiac cells.

## Figures and Tables

**Figure 1 genes-14-01328-f001:**
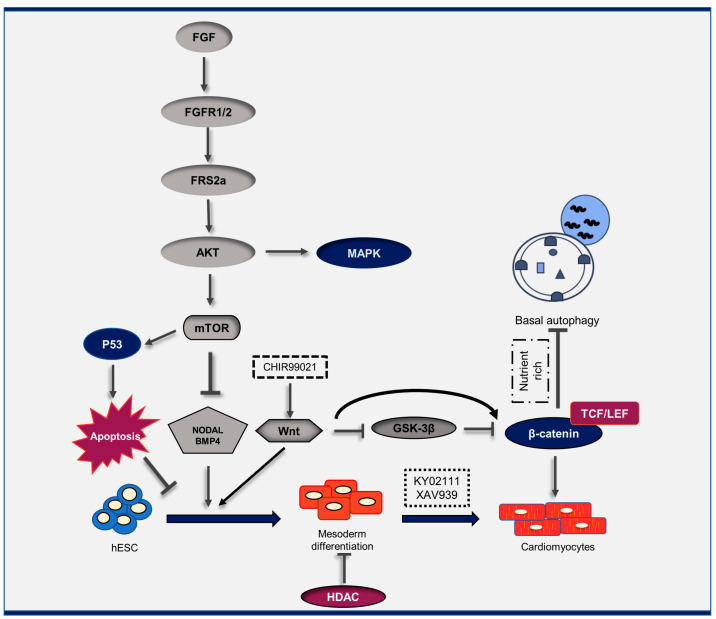
Schematic representation of the cross-talk between autophagy, cardiac differentiation, and other signaling pathways. Cardiac differentiation is regulated by a cascade of signaling pathways, such as autophagy, Wnt, Notch, and apoptosis. CHIR99021, a chemical compound that acts as an inhibitor of the enzyme GSK-3; XAV939, a Wnt/β-catenin inhibitor; KY02111, a Wnt/β-catenin inhibitor.

**Figure 2 genes-14-01328-f002:**
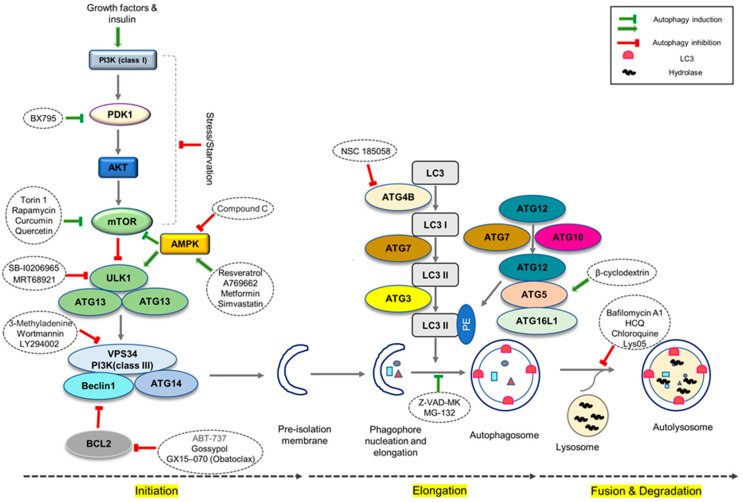
Autophagy modulators. Chemical inhibitors/inducers of the autophagy pathway and their targets.

**Figure 3 genes-14-01328-f003:**
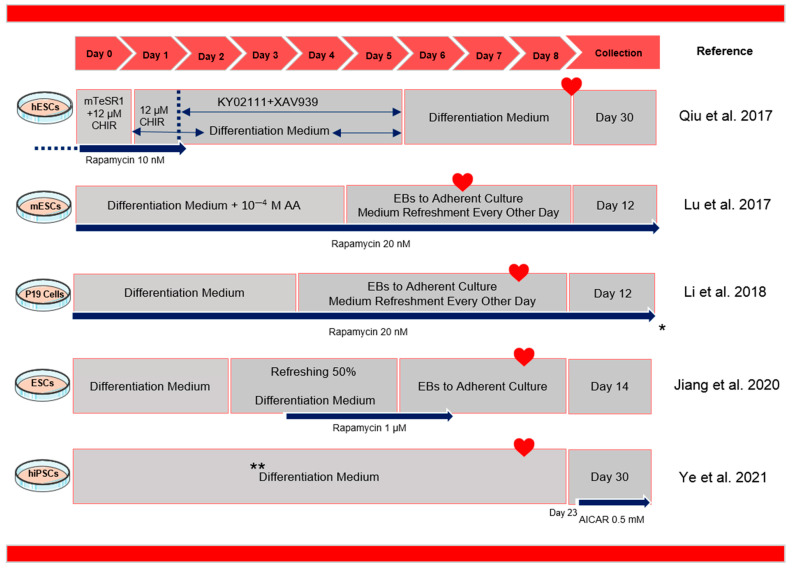
Protocols for the differentiation of cardiomyocytes based on the manipulation of the autophagy pathway. * The exact treatment days were not mentioned; ** The protocol days do not match the other protocols, for detailed information refer to the main reference [1,16,19,21,22]. Heart sign indicates the day beating started. The arrow sign indicated the start day of the treatment.

**Table 1 genes-14-01328-t001:** The role of autophagy manipulation in cardiac differentiation.

No.	Organism/Cell	Autophagy Inducer/Inhibitor	Genetic Model/Modification	Key Result	In Vivo/In Vitro	Ref.
1	Mouse	-	yes	FGF signaling promotes cardiac lineage determination at early stages and suppresses premature differentiation at late stages. In addition, autophagy plays a crucial role in mediating growth factor signaling pathways in regulating heart progenitor differentiation.	In vivo/In vitro	[62]
2	P19CL6 cell line	Rapamycin 3-MA MG-132	yes	The main effectors of Wnt and Notch signaling pathways, β-catenin and NICD, could form a complex with LC3 and P62 and could then be degraded by autophagy	In vitro	[37]
3	H9c2 cells SH-SY5Y cells	bafilomycin A1	no	During differentiation, HIF1α was upregulated and was required for cardiomyoblast differentiation as well as mitophagy.	In vitro	[55]
4	mES cells (D3)		shRNA-Raptor or shRNA-Rictor	Knockdown of both Raptor- and Rictor-suppressed mES cell differentiation into cardiomyocytes, which was similar to the effect of shRNA-Rictor in mES cells.	In vitro	[41]
5	hiPSCs	Rapamycin/NH4Cl	-	Rapamycin promoted EB-based cardiomyocyte differentiation of hiPSCs in a stage-dependent manner.	In vitro	[21]
6	mESCs	Rapamycin/Ku0063794/AA	-	Potential role of mTOR signaling in cardiac differentiation	In vitro	[1]
7	hESC (H9, H7)/hiPSCs (hiPS-U-Q1)	LiCl, HN4Cl, rapamycin, LY294002, Wortmannin, PD98059, PD0325901, SB431542, SB203580, SP600125, retinoic acid, Asiatic acid, Y27632, thiazovivin, z-VAD-FMK, VPA, TSA, VO-OHpic, SF1670, KU-55933, resveratrol, STR1720, CX-4945, ABT-737, nutlin-3, pifithrin-a, pifithrin-l, GSK1904529A, and FG-4592.	-	mTOR possesses a wide variety of effects on cardiogenesis from hPSCs	In vitro	[22]
8	hiPSCs	AICAR	-	Activation of AMPK by AICAR is an effective means to promote morphological and metabolic maturation of hiPSC-CMs		[19]
9	P19	H_2_O_2_, Rapamycin, Tunicamycin	-	The activation of Gata4 in transcription is promoted by appropriate ROS	In vitro	[16]
10	hiPSC (A18945)	-	CRISPR/Cas9-mediated knock out of Bcl-2	Loss of Bcl-2 delays cardiomyocyte development from hiPSC by regulating c-Myc expression	In vitro	[64]
11	hiPSC	Glucose starvation	Mfn2 shRNA PGAM5 shRNA	Mitophagy could help generate mature, differentiated cells	In vitro	[65]
12	Cardiac Sca-1+ cells from adult C57BL/6 mice	β-cyclodextrin (β-CD), 3-methyladenine (3MA), Bafilomycin A1 (Baf A1)	Atg5 knock down by siRNA	β-CD treatment induced autophagy differentiation through the JNK/STAT3 pathway	In vivo	[57]
13	hESCs	Metformin, HCQ	-	While the induction of autophagy leads to the production of mature cardiomyocytes, the inhibition of autophagy leads to a dramatic decrease in the number of cardiac cells.	In vitro	[17]

## Data Availability

Not applicable.

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
