# Peer review of "Revisiting the Role of Autophagy in Cardiac Differentiation: A Comprehensive Review of Interplay with Other Signaling Pathways"

_genes, 2023, doi:10.3390/genes14071328_

Round 1

Reviewer 1 Report

Lines 62-63: The authors state that modulation of autophagy may have beneficial effects. I recommend that they expand on this. What are the beneficial effects they are referring to?

Lines 65-68: The question that the authors pose is not entirely clear. I recommend that the authors re-structure this statement as to make it clear what is the object of this review article. What does enhancing the differentiation process of cardiomyocytes mean? Isn’t question already answered by the statement in lines 62-63?

Lines 71-73: The authors refer to the Zhang et al. manuscript, however, they do not go into specifics. In what way does FGF regulate cardiac development through autophagy? This information needs  to be provided.

Lines 77-79: How is the effect of Wnt signaling on autophagy affecting cardiac differentiation?

Lines 82-84: It is unclear what is the correlation between autophagy and cardiac differentiation.

Lines 92-95: What do the authors mean when they state that cardiomyocyte differentiation was increased?

Paragraph 89-108: Mechanistically, it is a bit unclear how these factors affect cardiogenesis. Is it due to increased mesoderm formation, increased differentiation of mesoderm into cardiomyocytes, increased activity of some transcription factor? The pathways and cellular states that are active/present during differentiation of ES or iPS cells into cardiomyocytes are well characterized (i.e., transcription factors, etc.). What the authors fail to do is to describe the connection between the main players of autophagy and the cardiac differentiation pathways or cellular states. For example, in line 100 the authors state “Rictor/mTORC2 plays a greater role in regulating cardiomyocyte differentiation from mESCs”. What does “greater role” mean? I recommend the authors become more granular with their explanations.

Paragraph 109-119: The relationship between mitophagy and cardiac differentiation is not clearly stated. The only definitive statement is “HIF upregulation during differentiation is required for cardiomyoblast development and mitophagy”. The authors fail to establish the relationship between mitophagy and cardiogenesis.

Paragraph 176-200: I fail to see the relationship between AMPK activity, cardiogenesis, and autophagy. 

Author Response

Dear Reviewer

Journal of Genes

Dear Reviewer,

We are grateful for your time and constructive feedback from the anonymous reviewers on our manuscript entitled “Revisiting the Role of Autophagy in Cardiac Differentiation: A Comprehensive Review of Interplay with Other Signaling Pathways” that we initially submitted under the manuscript number genes-2440015.

Please find our point-by-point response to your comments as below. All modifications made in the revised version of the manuscript are highlighted in yellow and track changed.

We would like to express our gratitude for considering the revised version of our manuscript and look forward to receive your decision.

Yours Sincerely,

Brunella franco, Ph.D.

Telethon Institute of Genetics and Medicine (TIGEM), Via Campi Flegrei, 34, 80078, Pozzuoli, Naples, Italy

First, to meet the minimum word limit requirement, we expanded the entire article with additional information and elaboration. Second, we have to thank for your insightful comments. We tried to address the comments and revise the manuscript accordingly.

Comment 1: Lines 62-63: The authors state that modulation of autophagy may have beneficial effects. I recommend that they expand on this. What are the beneficial effects they are referring to?

Answer to comment 1: According to the request of the respected reviewer, we tried to re-structure the mentioned part and changed it as below starting from line 59:

“In addition to autophagy, mitophagy, the selective degradation of dysfunctional or dam-aged mitochondria, plays a crucial role in cardiomyocyte differentiation by orchestrating a series of cellular processes. Mitochondria play an important role as key regulators in this developmental process because of their ability to respond to nutrient signals, dynamically adjust their respiratory capacity, and control the production of reactive oxygen species (ROS). Mitophagy is a process that allows cardiomyocytes to selectively eliminate dysfunctional mitochondria, ensuring the maintenance of a healthy mitochondrial population. This active mitochondrial quality control mechanism not only protects the cell from the deleterious effects of damaged mitochondria, but it also promotes the initiation of transcriptional and epigenetic changes that direct the maturation of cardiomyocytes [18].

Considering the benefits of regulating autophagy during the development of cardiomyocytes such as (but not limited to) improved cellular homeostasis, better control of protein quality, reduced oxidative stress, enhanced cell survival, and lower amount of pathological remodeling Numerous studies have indicated that modulating autophagy during cardiomyocyte development can have beneficial effects [1, 19-22], it becomes important to find out how modifying autophagy affects cardiomyocyte differentiation. Therefore, it is essential to investigate the impact of autophagy modification on cardiomyocyte differentiation. Additionally, establishing a reliable method for generating fully mature cardiomyocytes has been a persistent challenge with obvious translational implications, and autophagy modification may provide a potential solution. Therefore, the objective of this article is to summarize and review our knowledge in this field aiming at determining whether altering the autophagy pathway can enhance the differentiation process of cardiomyocytes and address this challenge.”

Comment 2: The question that the authors pose is not entirely clear. I recommend that the authors re-structure this statement as to make it clear what is the object of this review article. What does enhance the differentiation process of cardiomyocytes mean? Isn’t question already answered by the statement in lines 62-63

Answer to comment 2:  As answered in comment one, we tried to re-structure the whole paragraph, so this comment is also included and has been changed as follow starting from line 72:

“Considering the benefits of regulating autophagy during the development of cardiomyocytes such as (but not limited to) improved cellular homeostasis, better control of protein quality, reduced oxidative stress, enhanced cell survival, and lower amount of pathological remodeling Numerous studies have indicated that modulating autophagy during cardiomyocyte development can have beneficial effects [1, 19-22], it becomes important to find out how modifying autophagy affects cardiomyocyte differentiation. Therefore, it is essential to investigate the impact of autophagy modification on cardiomyocyte differentiation. Additionally, establishing a reliable method for generating fully mature cardiomyocytes has been a persistent challenge with obvious translational implications, and autophagy modification may provide a potential solution. Therefore, the objective of this article is to summarize and review our knowledge in this field aiming at determining whether altering the autophagy pathway can enhance the differentiation process of cardiomyocytes and address this challenge.”

Comment 3: Lines 71-73: The authors refer to the Zhang et al. manuscript, however, they do not go into specifics. In what way does FGF regulate cardiac development through autophagy? This information needs to be provided.

Answer to comment 3: The comment 3 is answered as follow in the manuscript starting from line 88:

“The crosstalk between autophagy and other signaling pathways plays a crucial role in the modulation of cardiac differentiation. Zhang et al. were the first to recognize the connection between autophagy and cardiac stem cell differentiation, finding that fibroblast growth factor (FGF) regulates cardiac development through autophagy [8]. Their study revealed that the inhibition of FGF signaling led to an upsurge in the initiation of autophagy, thereby promoting the differentiation of cardiomyocytes. These findings strongly suggest that autophagy serves as a positive regulator of both cardiac progenitor cell differentiation and cardiomyocyte maturation.

FGF signaling exhibits a biphasic function in cardiogenesis. During the early stages, FGF signaling promotes mesoderm formation and commitment to the cardiac lineage. How-ever, in later stages, FGF signaling inhibits premature differentiation to ensure proper cardiac development and maturation. The involvement of autophagy in mediating FGF signaling during the differentiation of heart progenitor cells highlights the significance of autophagy in cardiac development. The interplay between autophagy and FGF signaling provides novel insights into the regulatory mechanisms underlying cardiac differentiation. Moreover, FGF signaling has been found to inhibit autophagy activity, while the suppression of FGF signaling increases autophagy initiation [8, 10]. This further supports the intricate relationship between FGF and autophagy in cardiac differentiation.”

Comment 4: Lines 77-79: How is the effect of Wnt signaling on autophagy affecting cardiac differentiation?

Answer to comment 4: The comment 4 is answered as follow in the manuscript starting from line 106:

“Additionally, other signaling pathways, such as Wnt, Notch, and bone morphogenetic proteins (BMPs), are also involved in regulating heart development [23]. Wnt and Notch signaling pathways are important in heart development, and there is evidence of a cross-talk between the two [24]. Main components of the Wnt signalling pathway include Dishevelled (Dvl), AXIN, glycogen synthase kinase-3 (GSK3)-β, and β-catenin. It has been shown that Wnt signaling plays a biphasic function in cardiogenesis. This pathway stimulates cardiomyocyte differentiation in the early stages of heart development, but sup-presses cardiogenesis later on [25]. Autophagy can be negatively regulated by Wnt signaling by promoting Dvl degradation through the autophagy-lysosome pathway, which in turn induces a concomitant decrease in the nuclear accumulation of β-catenin [26]. Furthermore, it has been demonstrated that autophagy induction resulted in a considerable drop in the gene expression of Axin2, Cyclin-D1, and c-Myc, as well as a decrease in the protein expression of β-catenin and Dvl2. Based on these findings, it was determined that autophagy has a negative regulatory effect on the Wnt signaling pathway [1]. Moreover, GSK3, a component of the β-catenin-WNT pathway, directly regulates autophagy through the GSK3-KAT5/TIP60-Unc-51 like autophagy activating kinase 1 (ULK1) signal [27]. Additionally, the differentiation of cardiac stem cells (CSCs) into cardiomyocytes is influenced by the Jun N-terminal kinase (JNK) / signal transducer and activator of transcription 3 (STAT3) and β-catenin pathways [28-31].”

Comment 5: Lines 82-84: It is unclear what is the correlation between autophagy and cardiac differentiation.

Answer to comment 5: The comment 5 is answered as follow in the manuscript starting from line 136 and the whole paragraph tried to be changed to convey the information better:

“Notch signaling, a pivotal signaling pathway in cardiac differentiation, demonstrates intricate dynamics and complexity throughout the process of cardiac development. Initially, it is believed that Notch signaling suppresses cardiac differentiation to maintain a pool of undifferentiated progenitor cells. However, different studies show that Notch1 is required for cardiac progenitor cells to differentiate into cardiomyocytes by positively regulating the expression of cardiac transcription factors in mouse embryos and embryonic stem (ES) cells [32-34]. In the context of cardiac development, autophagy has been investigated in relation to the Wnt and Notch signaling pathways [35, 36]. Zhuqing Jia et al. studied the involvement of autophagy in the cardiac development of P19CL6 cells, a well-established in vitro cardiomyocyte differentiation system [37]. Their findings demonstrated that autophagy is initiated during the early stages of cardiomyocyte differentiation and remains active throughout the final stages. The researchers also observed that β-catenin and Notch intracellular domain (NICD), which are effectors of the Wnt and Notch pathways, can form a complex with LC3 and P62, leading to their elimination through autophagy. Interestingly, β-catenin not only inhibits the formation of autophagosomes but also directly reduces the expression of the autophagy adaptor p62 [36]. Because both the Notch and Wnt pathways are inhibitory in the late stages of cardiac differentiation, appropriate clearance of both pathways' components may allow P19CL6 cells to complete the cardiac differentiation process [37, 38].

Moreover, epigenetic factors like histone deacetylases (HDACs), have also been shown to play a role in regulating autophagy during cardiac differentiation [27-31]. Specifically, studies have demonstrated that HDACs control the differentiation of P19 embryonic cancer cells into cardiac lineage cells [39]. Despite the advances in understanding the roles of autophagy and epigenetic regulation in cardiac differentiation, the exact mechanisms underlying their interaction in this process remain incompletely understood. Further investigations are required to elucidate the intricate relationship between autophagy, Wnt signaling, and cardiac differentiation, which will provide valuable insights into the development and potential therapeutic strategies for cardiac-related conditions.”

Comment 6: Reviewer 1: Mechanistically, it is a bit unclear how these factors affect cardiogenesis. Is it due to increased mesoderm formation, increased differentiation of mesoderm into cardiomyocytes, increased activity of some transcription factor? The pathways and cellular states that are active/present during differentiation of ES or iPS cells into cardiomyocytes are well characterized (i.e., transcription factors, etc.). What the authors fail to do is to describe the connection between the main players of autophagy and the cardiac differentiation pathways or cellular states. For example, in line 100 the authors state “Rictor/mTORC2 plays a greater role in regulating cardiomyocyte differentiation from mESCs”. What does “greater role” mean? I recommend the authors become more granular with their explanations.

Answer to comment 6: The comment 6 is answered as follow in the manuscript starting from line 164: “The mammalian target of rapamycin (mTOR) is a crucial serine/threonine kinase involved in cellular survival and growth. It operates through two distinct complexes, mTOR com-plex 1 (mTORC1) and mTOR complex 2 (mTORC2), which are predominantly made up of Raptor and Rictor as their core components, respectively. Reports from different laboratories have demonstrated that mTOR complex 1 acts as a master regulator of autophagic processes since inhibition of mTORC1 was shown to be required to start autophagy. Sub-sequent studies have also demonstrated that mTORC1 directly regulates the subsequent phases such as autophagosomes formation, maturation and autophagy termination [40]. Raptor/mTORC1 and Rictor/mTORC2 play distinct and significant roles in the differentiation of cardiomyocytes derived from mouse embryonic stem cells (mESCs). Numerous molecular elements and phosphorylation processes mediate the intricate interaction between these two signaling pathways [41].

AKT, also known as protein kinase B, is a key protein and acts as a bridge between the two complexes. AKT can be phosphorylated by mTORC2 or 3-phosphoinositide-dependent kinase 1 (PDK1), thereby regulating downstream signaling [42]. Specifically, Raptor knockdown reduced the phosphorylation of Rictor at Thr1135 by p70S6K, leading to mTORC2 activation [41].

Raptor/mTORC1 primarily regulates metabolism, cell growth, and protein synthesis in response to nutrient availability and growth factors [43]. It controls the phosphorylation of S6 kinase 1 (S6K1) and eukaryotic translation initiation factor 4E-binding protein (eIF4E), thereby promoting translation and protein synthesis [44]. In the context of cardiomyocyte differentiation, Raptor knockdown enhances the process by activating Rictor/mTORC2 signaling, which in turn facilitates the differentiation of mESCs into cardiomyocytes. This accompanied by elevated expression levels of brachyury (a mesoderm protein), Nkx2.5 (a cardiac progenitor cell protein), and α-actinin (a cardiomyocyte marker) in Raptor knock-down cells [41].

On the other hand, Rictor/mTORC2 exerts its impact on cytoskeletal organization, cell survival, and cell polarization. It has been implicated in the regulation of cellular processes such as cell adhesion, migration, and differentiation [45]. Rictor knockout in cardiac-specific cells leads to cardiac dysfunction and impaired adaptation to pressure over-load. In the context of mESCs differentiation into cardiomyocytes, Rictor knockdown sup-presses cardiogenesis, indicating the critical role of Rictor/mTORC2 in this process [41].

Upstream of mTOR, various signals, including insulin signaling through phosphatidyl-inositol-3-kinase (PI3K) and AKT, and energy/stress response signaling via AMP-activated protein kinase (AMPK), converge to regulate the activity of the tuberous sclerosis complexes (TSC1-TSC2), which act as GTPase-activating proteins for Rheb. When Rheb is in its GTP-bound state, it directly activates mTOR. The convergence of these signals ensures precise control over mTOR activation and downstream cellular processes [44].

The regulation of cardiomyocyte differentiation can be better understood by gaining valuable insights from the complex relationship between these signaling pathways, which may also have implications for potential therapeutic applications using induced pluripotent stem cells.”

Comment 7: Lines 92-95: What do the authors mean when they state that cardiomyocyte differentiation was increased?

Answer to comment 7: To answer this question following sentences were added to the manuscript, starting from line 190: “In the context of cardiomyocyte differentiation, Raptor knockdown enhances the process by activating Rictor/mTORC2 signaling, which in turn facilitates the differentiation of mESCs into cardiomyocytes. This accompanied by elevated expression levels of brachyury (a mesoderm protein), Nkx2.5 (a cardiac progenitor cell protein), and α-actinin (a cardio-myocyte marker) in Raptor knockdown cells [41].”

Comment 8: Paragraph 109-119: The relationship between mitophagy and cardiac differentiation is not clearly stated. The only definitive statement is “HIF upregulation during differentiation is required for cardiomyoblast development and mitophagy”. The authors fail to establish the relationship between mitophagy and cardiogenesis.

Answer to comment 8: To answer this question following sentences were added to the manuscript, starting from line 216: “Aside from the main autophagy pathway, other types of autophagy, such as mitophagy, play a crucial role in cardiac differentiation. Mitophagy, a form of autophagy, is a vital process in cardiac differentiation. It helps in the removal of dysfunctional mitochondria and facilitates metabolic remodeling, which is crucial for successful differentiation. Regulated mitophagy plays a critical role in maintaining the cell's glycolytic status and is believed to contribute to the metabolic remodeling necessary for successful differentiation [46-49].

A specific process of mitophagy is initiated by the PINK1/Parkin system, where Parkin, an E3 ubiquitin ligase, is recruited to dysfunctional mitochondria. Parkin ubiquitinates outer mitochondrial proteins, labeling them for proteasome degradation [50]. In mice, the deletion of Parkin during early postnatal stages impedes cardiomyocyte maturation, affects mitochondrial biogenesis, and disrupts fatty acid oxidation, highlighting the importance of PINK1/Parkin-mediated mitophagy in cardiac development [51]. These findings high-light the important role of PINK1/Parkin-mediated mitophagy in promoting proper cardiac development.

The regulation of mitophagy in cardiac differentiation is influenced by nutrient sensing pathways such as Akt/mTOR, which is upregulated in high-nutrient states, and AMPK, which is upregulated during fasting. Inhibiting mTOR signaling increases mitophagy, providing cardio protection after myocardial infarction. Conversely, AMPKα2 phosphorylates PINK1, enhancing mitophagy and protecting against pressure overload in the heart. AMPK also influences the opening of the mitochondrial permeability transition pore (mPTP), which regulates mitochondrial depolarization and subsequent mitophagy [18, 52]. Increasing Parkin-mediated mitophagy through enhanced fatty acid oxidation, achieved by deleting acetyl coenzyme A carboxylase 2 (ACC2), prevents myocardial dysfunction resulting from a high-fat diet [53]. These interconnected mechanisms underscore the significance of metabolic shifts, mitochondrial quality control, and mitophagy in maintaining proper cardiac function.

Conversely, inhibiting mitophagy by chronically activating monoamine oxidase-A (MAO-A) leads to the accumulation of dysfunctional mitochondria, promoting a senescent phenotype. This inhibition upregulates cell cycle inhibitors, triggers the ROS-induced DNA damage response, and increases senescence-associated β-galactosidase (SA-β-gal) activity. Hence, maintaining mitophagy is crucial for inhibiting senescence, promoting cardiomyocyte maturation, removing dysfunctional mitochondria, and supporting optimal mitochondrial function [54].

The hypoxia-inducible factor 1 (HIF-1) is one of the key regulators of mitophagy during cardiac differentiation. HIF-1 is known to stabilize and activate in response to low oxygen levels, and it has been found to play a critical role in cardiomyocyte development [55]. Zhao et al. (2020) found that HIF-1 is upregulated during differentiation and triggered NIX (also known as BNIP3L)-dependent mitophagy [56]. NIX acts as a mitophagy receptor that facilitates the selective engulfment of mitochondria by autophagosomes. Activation of HIF-1 during cardiomyocyte differentiation leads to the stabilization of NIX, promoting the removal of dysfunctional mitochondria through mitophagy. This controlled elimination of damaged mitochondria is essential for maintaining the overall health and functionality of differentiating cardiomyocytes. Mitophagy helps eliminate dysfunctional mitochondria and may also contribute to metabolic remodeling essential for differentiation [39-42]. Using a cardiomyocyte differentiation cell line model, Zhao et al. found that hypoxia-inducible factor (HIF)-1 stabilized during differentiation and triggered NIX (also known as BNIP3L) -dependent mitophagy [43]. Controlled mitophagy is thought to maintain or improve the cell's glycolytic status, potentially through a positive feedback loop with HIF1, although the mechanism is not yet understood. Changing NIX expression levels can affect the rate of differentiation, and HIF1 upregulation during differentiation is required for cardiomyoblast development and mitophagy [56, 57].”

Comment 9: I fail to see the relationship between AMPK activity, cardiogenesis, and autophagy.

Answer to comment 9: To answer this question following sentences were added to the manuscript, starting from line 327: “AMPK is an energy-sensing protein kinase involved in the regulation of autophagy. In addition to its direct role in regulating autophagy, AMPK can also influence other cellular processes, such as mitochondrial function, posttranslational acetylation, cardiomyocyte metabolism, mitochondrial autophagy, endoplasmic reticulum stress, and apoptosis. As AMPK is involved in the control of various cellular processes, it can influence the health and survival of cardiomyocytes [20, 66]. AMPK activation increases fatty acid oxidation in human induced pluripotent stem cell (hiPSC)-cardiomyocytes (CMs) by upregulating the expression of carnitine palmitoyl transferase (CPT)-1, fatty acid transport protein (FATP), and fatty acid-binding protein (FABP) [20, 67]. Moreover, AMPK has a distinct regulatory impact on numerous areas of mitochondrial biology and homeostasis, such as controlling the number of mitochondria via stimulating mitochondrial synthesis, regulating the architecture of the mitochondrial network, and controlling the quality of mitochondria via autophagy and mitosis [68].

Understanding the significance of AMPK in cardiomyocyte differentiation, Ye et al. employed 5-aminoimidazole-4-carboxamide-1-β-D-ribofuranoside (AICAR) to induce AMPK activation in hiPSC-CMs [19]. AICAR dramatically raised the oxygen consumption rate (OCR) related with baseline respiration, adenosine 5′-triphosphate (ATP) generation, maximum respiration, and reserve capacity, indicating that cardiomyocytes were in a more mature metabolic state. Furthermore, AMPK activation enhances mitochondrial fusion in hiPSC-CMs [19]. In times of cellular stress, mitochondrial respiratory reserve capacity is employed to sustain an increase in energy demand, assisting in the maintenance of cell and organ functions, cell repair, or active drug detoxification [69]. AICAR treatment boosted respiratory reserve capacity in hiPSC-CMs, implying that the treated cells may perform better in the face of increasing energy demand. Also, AICAR-treated hiPSC-CMs exhibited a larger cell perimeter, a lower circularity index, and longer sarcomeres than the control group making them more similar to adult cardiomyocytes [19]. AICAR also increased the expression of cardiac maturation-related genes. Although the AICAR activation of AMPK promotes morphological and metabolic maturity of hiPSC-CMs, the effect of AMPK on electrophysiological maturation of hiPSC-CMs remains unknown [19].

In a recent study, the impact of an AMPK inducer (Metformin) on the differentiation of hPSC-CMs was investigated. The findings demonstrated a significant upregulation of autophagy during cardiac differentiation following AMPK activation. Additionally, this autophagy upregulation was associated with an increase in the expression of CM-specific markers in hPSC-CMs. These findings provide valuable insights into the vital role of autophagy in the process of cardiomyocyte differentiation [17].”

We hope these revisions address your concerns and that our manuscript is now suitable for publication. We are looking forward to your positive response and further guidance.

Reviewer 2 Report

It is interesting to review the role of autophagy in cardiac differentiation.

But this review was not well organized and it is not so easy to get key points.

In Figure 2, the authors draw the chemical inhibitors/inducers of the autophagy pathway and their targets. It is better to link them with the cardiac differentiation.

It's good writing.

Author Response

Dear Reviewer

Journal of Genes

Dear Reviewer,

We are grateful for your time and constructive feedback from the anonymous reviewers on our manuscript entitled “Revisiting the Role of Autophagy in Cardiac Differentiation: A Comprehensive Review of Interplay with Other Signaling Pathways” that we initially submitted under the manuscript number genes-2440015.

Please find our point-by-point response to your comments as below. All modifications made in the revised version of the manuscript are highlighted in yellow and track changed.

We would like to express our gratitude for considering the revised version of our manuscript and look forward to receive your decision.

Yours Sincerely,

Brunella franco, Ph.D.

Telethon Institute of Genetics and Medicine (TIGEM), Via Campi Flegrei, 34, 80078, Pozzuoli, Naples, Italy

First, to meet the minimum word limit requirement, we expanded the entire article with additional information and elaboration. Second, we have to thank you for your insightful comments. We tried to address the comments and revise the manuscript accordingly.

Comment 1: It is interesting to review the role of autophagy in cardiac differentiation. But this review was not well organized and it is not so easy to get key points.

Answer to comment 1: We greatly appreciate your valuable feedback on our manuscript. We have thoroughly revised and reorganized the content to enhance clarity and ensure that the key points are easily discernible.

Comment 2: In Figure 2, the authors draw the chemical inhibitors/inducers of the autophagy pathway and their targets. It is better to link them with the cardiac differentiation.

Answer to comment 2: We appreciate the referee's suggestion to link the inhibitors and inducers of autophagy in Figure 2 with cardiac differentiation. We understand the importance of providing a clear connection between these factors for a better understanding of their implications in the context of cardiac differentiation.

In response to the referee's comment, we would like to clarify the specific objectives of Figure 2. While Figure 1 aims to illustrate the crosstalk between autophagy and cardiac differentiation based on available knowledge, Figure 2 has a distinct focus on presenting a comprehensive overview of the inhibitors and inducers of autophagy. The intention behind Figure 2 is to provide researchers and readers with a quick reference guide, allowing them to identify the chemicals that target the autophagy pathway.

We hope these revisions address your concerns and that our manuscript is now suitable for publication. We are looking forward to your positive response and further guidance.

Round 2

Reviewer 1 Report

The authors have adequately addressed the comments that I have raised. The review manuscript is in good shape for publication. 

No comment